# Agnostic Multi-Group Active Learning

**Nick Rittler**
University of California - San Diego
nrittler@ucsd.edu

**Kamalika Chaudhuri**
University of California - San Diego
kamalika@cs.ucsd.edu

## Abstract

Inspired by the problem of improving classification accuracy on rare or hard subsets of a population, there has been recent interest in models of learning where the goal is to generalize to a collection of distributions, each representing a "group". We consider a variant of this problem from the perspective of active learning, where the learner is endowed with the power to decide which examples are labeled from each distribution in the collection, and the goal is to minimize the number of label queries while maintaining PAC-learning guarantees. Our main challenge is that standard active learning techniques such as disagreement-based active learning do not directly apply to the multi-group learning objective. We modify existing algorithms to provide a consistent active learning algorithm for an agnostic formulation of multi-group learning, which given a collection of $G$ distributions and a hypothesis class $\mathcal{H}$ with VC-dimension $d$, outputs an $\epsilon$-optimal hypothesis using $\tilde{O}\left((\nu^2/\epsilon^2)Gd\theta_{\mathcal{G}}^2 \log^2(1/\epsilon) + G\log(1/\epsilon)/\epsilon^2\right)$ label queries, where $\theta_{\mathcal{G}}$ is the worst-case disagreement coefficient over the collection. Roughly speaking, this guarantee improves upon the label complexity of standard multi-group learning in regimes where disagreement-based active learning algorithms may be expected to succeed, and the number of groups is not too large. We also consider the special case where each distribution in the collection is individually realizable with respect to $\mathcal{H}$, and demonstrate $\tilde{O}\left(Gd\theta_{\mathcal{G}} \log(1/\epsilon)\right)$ label queries are sufficient for learning in this case. We further give an approximation result for the full agnostic case inspired by the group realizable strategy.

## 1  Introduction

There is a growing theory literature concerned with choosing a classifier that performs well on multiple subpopulations or "groups" [1, 2, 7, 6, 4, 5, 6, 3]. In many cases, the motivation comes from a perspective of fairness, where a typical requirement is that we classify with similar accuracy across groups [7, 6, 4]. In other cases, the motivation may simply be to train more reliable classifiers. For example, cancer detection models with good overall accuracy often suffer from poor ability to detect rare subtypes of cancer that are not well-represented or identified in training. This suggests that naive ERM may be insufficient in practice [8].

In this work, we consider the following formulation of "multi-group" learning. The learner is given a collection of distributions $\mathcal{G} = \{D_g\}_{g=1}^{G}$, each corresponding to a group, and a hypothesis class $\mathcal{H}$, and wants to pick a classifier that approximately minimizes the maximum classification error over group distributions. We consider this problem from an active learning perspective, where the learner has the power to choose which examples from each group it wants to label during training. In a standard extension of the active learning literature, we set out to design schemes for choosing which examples from each group should be labeled, where the goal is to minimize the number of label queries while retaining PAC-learning guarantees.

A major challenge in harnessing the power of active algorithms even in standard agnostic settings is making sure they are consistent. In the case of active learning, this means that as the number

37th Conference on Neural Information Processing Systems (NeurIPS 2023).

of number of labels requested approaches infinity, the learner outputs an optimal hypothesis. To complicate things further, the main algorithmic paradigm for consistent agnostic active learning over a single distribution - disagreement based active learning (DBAL) - fails to admit direct application in the multi-group learning objective. The fundamental idea in DBAL is that the learner may safely spend its labeling budget in the "disagreement region", a subset of instance space where empirically well-performing hypotheses disagree about how new examples should be labeled. When the learner need only consider a single distribution, error differences between classifiers are specified entirely through their performance on the disagreement region, and so spending the labeling budget here allows the learner to figure out which hypotheses are best while saving on labels. The problem is that when multiple group distributions must be considered, the absolute errors of classifiers on each group must be estimated to compare performance of two classifiers in their worst case over collection, and this property no longer holds.

We resolve this via the observation that, while we cannot spend all our labeling budget in the disagreement region, we can exploit the agreement in its complement to cheaply estimate absolute errors of classifiers on each group. In particular, we estimate the absolute errors by choosing a representative classifier $h_{\mathcal{H}'}$ in the set of empirically well-performing classifiers $\mathcal{H}'$, and estimating its error on the complement of the disagreement region on each group distribution. These error estimates can be used to construct estimates for the absolute errors on each group for each $h \in \mathcal{H}'$ at the statistical cost of estimating a coin bias, leading to a relatively cheap, consistent active strategy.

We analyze the number of label queries made by this scheme in terms of a standard complexity measure in the active learning literature called the "disagreement coefficient" [9, 10], and show an upper bound of $\tilde{O}\left((\nu^2/\epsilon^2)Gd\theta_{\mathcal{G}}^2\log^2(1/\epsilon) + G\log(1/\epsilon)/\epsilon^2\right)$, where $\theta_{\mathcal{G}}$ is the maximal disagreement coefficient over the collection of group distributions. We discuss some regimes where this label complexity can be expected to push below sample complexity lower bounds for a learner that can request samples from each group distribution during training, but does not have power to abstain from labeling specific examples.

We also consider the special case of agnostic learning where each group distribution is individually realizable, but no single hypothesis separates all groups simultaneously. In this case, we show that all dependence on $1/\epsilon^2$ in the label complexity can be replaced with $\log(1/\epsilon)$ when disagreement coefficients are bounded. It turns out that using the strategy we develop in this special case leads to an approximation algorithm for the general agnostic case, for which we give guarantees.

## 2   Related Work

### 2.1   Multi-Group Learning

The majority of the empirical work on multi-group learning has been through the lens of "Group-Distributionally Robust Optimization" (G-DRO) [11, 12, 13]. The goal in G-DRO is to choose a classifier that minimizes the maximal risk against an unknown mixture over a collection of distributions $\{D_g\}_{g=1}^G$ representing groups. One assumes a completely passive sampling setting - all data is given to the learner at the beginning of training, and the learner has no ability to draw extra, fine-grained samples. The strategy is usually empirical risk minimization (ERM) - or some regularized variant - on the empirical max loss over groups; for a set of classifiers parameterized by $\phi \in \Phi$, letting $S_g$ denote the set of examples in the training set coming from $D_g$, one performs $\min_{\phi \in \Phi} \max_{g \in [G]} \frac{1}{|S_g|} \sum_{(x_i, y_i) \in S_g} l(f_\phi(x_i), y_i)$ for some loss $l$. It is important to note that the learner knows the group identity of each sample in the training set, but is not provided with group information at test time, precluding the possibility of training a separate classifiers for each group.

"Multi-group PAC learning" consider the multi-group problem under the passive sampling assumption from a more classical learning-theoretic perspective [7, 6]. Here, one assumes there is a single distribution $D$ from which one is given samples, but also a collection of subsets of instance space $\mathcal{G}$ over which one wants to learn conditional distributions. Given a hypothesis class $\mathcal{H}$, the learner tries to improperly learn a classifier $f$ that competes with the optimal hypothesis on each conditional distribution specified by a group $g$ in the collection - formally, one requires that for a given error tolerance $\epsilon$, $f$ has the property $\forall g \in \mathcal{G}, \ \mathbb{P}_D(f(x) \neq y | x \in g) \leq \inf_{h \in \mathcal{H}} \mathbb{P}_D(h(x) \neq y | x \in g) + \epsilon$ with high probability. An interesting wrinkle in this literature is that the group identity of samples is available at both training and test times. It has been shown that a sample complexity of

Table 1: Overview of the complexity of multi-group learning. The $\tilde{O}$ notation hide factors logarithmic in $d$, $G$, $1/\delta$, and $\log(1/\epsilon)$. We reserve discussion of regimes in which our algorithm improves on results in Collaborative Learning for Section 5.

| Problem | Full Agnostic | Group-Realizable |
|---|---|---|
| Passive Multi-Group [6] | $\tilde{O}\left(\log(|\mathcal{H}|)/\gamma\epsilon^2\right)$ | $\tilde{O}\left(\log(|\mathcal{H}|)/\gamma\epsilon\right)$ |
| Collaborative Learning [3] | $\tilde{O}\left(d\log(1/\epsilon)/\epsilon^2 + G/\epsilon^2\right)$ | ? |
| Active Multi-Group (us) | $\tilde{O}\left((\nu^2/\epsilon^2)Gd\theta_{\mathcal{G}}^2\log^2(1/\epsilon) + G\log(1/\epsilon)/\epsilon^2\right)$ | $\tilde{O}\left(Gd\log^2(1/\epsilon)\right)$ |

$\tilde{O}\left(\log(|\mathcal{H}|)/\gamma\epsilon^2\right)$ is sufficient for agnostic learning in this model, where $\gamma$ is the minimal mass of a group $g$ under $D$ [6].

"Collaborative learning" studies the multi-group problem under an alternative sampling model [1, 2, 3]. In this case, we are given a collection of distributions $\{D_g\}_{g=1}^G$, each corresponding to a group. Given some hypothesis class $\mathcal{H}$, the goal is to learn a classifier $f$, possibly improperly, that is evaluated against its worst-case loss over $D_1, \ldots, D_G$; formally, we would like $f$ to satisfy $\max_{g\in[G]} \mathbb{P}_{D_g}(f(x) \neq y) \leq \inf_{h\in\mathcal{H}} \max_{g\in[G]} \mathbb{P}_{D_g}(h(x) \neq y) + \epsilon$. In contrast with multi-group PAC learning, the learner may decide how many samples from each $D_g$ it wants to collect during training, and group identity is hidden on test examples. This models the case where a learner may want to collect more data from a particularly difficult group of instances, such as a rare or hard-to-diagnose type of cancer. It has been shown for finite hypothesis classes that $\tilde{\Theta}(\log(|\mathcal{H}|)/\epsilon^2 + G/\epsilon^2)$ total samples over all groups are necessary and sufficient to learn in this model; $\tilde{O}(d\log(1/\epsilon)/\epsilon^2 + G/\epsilon^2)$ total samples are sufficient for VC-classes [3].

Our work extends the model of collaborative learning, and endows the learner with the ability to decide which samples from each group distribution $D_g$ should be labeled. This is the standard framework of active learning, applied to the multi-group setting. As in collaborative learning, we assume group identity is hidden at test time.

## 2.2 Active Learning

Active learning concerns itself with the development of learning algorithms for training classifiers that have power over which training examples should be labeled [14, 15]. The field has largely focused on uncovering settings in which algorithmic approaches reduce the amount of labels required for PAC-style learning guarantees beyond sample complexity lower bounds that apply to i.i.d. data collection from the underlying distribution [16, 17]. In the agnostic, 0-1 loss setting, the standard upper bounds for label complexity follow $\tilde{O}\left(\theta(d\log(1/\epsilon) + d\nu^2/\epsilon^2)\right)$. Here, $\nu$ is the "noise rate", i.e. the true error of the optimal hypothesis $h^*$, and $\theta$ is a distribution-dependent parameter called the "disagreement coefficient". Thus, gains of active strategies over standard passive lower bounds of $\Omega(d\nu/\epsilon^2)$ depend on certain easiness conditions like small noise rates and bounded disagreement coefficients [15].

The vast majority of the work on active learning has been done in the 0-1 loss setting [18, 9, 10, 19, 20, 21]. It has been significantly harder to push the design of active learning algorithms past the regime of accuracy on a fixed distribution. While some work has attempted to generalize classical ideas of active learning to different losses [22], these are heavily outnumbered in the literature.

As previously mentioned, the most difficult part of designing agnostic active learning strategies is maintaining consistency. The issue comes down to a phenomenon referred to as "sampling bias" : because active learners would like to target certain parts of space to save on labels, there is a risk that the learner prematurely stops sampling on a part of space in which there is some detail in the distribution that could not be detected at a higher labeling resolution. This can easily lead to inconsistent strategies [15]. Thus, a major contribution of our work is exhibiting a consistent active scheme for the multi-group problem.

# 3 Preliminaries

## 3.1 Learning Problem

We study a binary classification setting where examples fall in some instance space $\mathcal{X}$, and labels lie in $\mathcal{Y} := \{-1, 1\}$. We suppose we are given some pre-specified, finite collection of distributions $\mathcal{G} = \{D_g\}_{g=1}^G$ over $\mathcal{X} \times \mathcal{Y}$ corresponding to groups. Given a hypothesis class $\mathcal{H}$ of measurable classifiers with VC-dimension $d$, the goal of the leaner is to pick some $h \in \mathcal{H}$ from finite data that performs well across all the distributions in $\mathcal{G}$ in the worst case. Let $L_\mathcal{G}(h \mid g) := \mathbb{P}_{D_g}(h(x) \neq y)$ be the error of a hypothesis $h$ on group $g$. Formally speaking, the learner would like to choose a classifier approximately obtaining

$$\inf_{h \in \mathcal{H}} \max_{g \in [G]} L_\mathcal{G}(h \mid g),$$

using finite data. We often use $L_\mathcal{G}^{\max}(h)$ as shorthand for $\max_{g \in [G]} L_\mathcal{G}(h \mid g)$. We use $\nu := \inf_{h \in \mathcal{H}} L_\mathcal{G}^{\max}(h)$ to denote the "noise rate" of $\mathcal{H}$ on the multi-distribution objective. The use of the term "agnostic" throughout reflects the fact that we make no assumption that $\nu = 0$ in our algorithm design or analysis. We assume for simplicity that there is some $h^* \in \mathcal{H}$ attaining $\nu$.

## 3.2 Active Learning Model

We consider a standard active learning model specified as follows. Let $supp(g)$ denote the support of the marginal over instance space of $D_g$. The active learner has access to two sampling oracles for each distribution specified by $D_g$. The first is $U_g(\cdot)$, which given a set $S \subseteq \mathcal{X}$ measurable with respect to $D_g$, returns an unlabeled sample from $D_g$ conditioned on $S$; if $\mathbb{P}_{D_g}(x \in S) = 0$, then $U_g(S)$ returns "None". The second is $O_g(\cdot)$, which given a point in $supp(g)$, returns a sample from the conditional distribution over labels specified by $x$ and $g$. More formally, querying $U_g(S)$ for $S$ such that $\mathbb{P}_{D_g}(x \in S) \neq 0$ is equivalent to drawing i.i.d. samples according to marginal over instance space of $D_g$ (independent of previous randomness), and returning the first example that falls in $S$; querying the oracle $O_g(x)$ for $x \in supp(g)$ is equivalent to receiving a sample from a Rademacher random variable with parameter $\mathbb{P}_{D_g}(Y = 1 | X = x)$.

As is standard in active learning, the active learner is assumed to have functionally unlimited access to queries from $U_g(\cdot)$. On the other hand, queries to oracles $O_g(\cdot)$ are precious: the "label complexity" of a strategy executed by the active learner is the sum of queries to oracles $O_g(\cdot)$ over all $g$, and is to be minimized given a desired generalization error guarantee.

# 4 Challenges in Multi-Group Active Learning

In this section, we give some background on classical disagreement-based methods on a single distribution, and discuss in more detail the challenge of designing consistent active learning strategies in the multi-group setting.

## 4.1 Background on Disagreement-Based Active Learning

Almost all agnostic active learning algorithms for accuracy over a single distribution boil down to disagreement-based methods [18, 10, 15, 16]. The fundamental idea in this school of algorithms is that one can learn the relative accuracy of two classifiers $h$ and $h'$ by only requesting labels for examples in the part of instance space on which they disagree about how examples should be labeled. More generally, given a set of classifiers $\mathcal{H}' \subseteq \mathcal{H}$, one can consider the "disagreement region" of $\mathcal{H}'$, defined as

$$\Delta(\mathcal{H}') := \{x \in \mathcal{X} : \exists h, h' \in \mathcal{H}' \ s.t. \ h(x) \neq h'(x)\}.$$

As alluded to above, the difference in accuracy of classifiers $h, h' \in \mathcal{H}'$ is specified entirely through this inherently label-independent notion. For a single distribution $D$, we may write

$$\frac{\mathbb{P}_D(h(x) \neq y) - \mathbb{P}_D(h'(x) \neq y)}{\mathbb{P}_D(\Delta(\mathcal{H}'))} = \mathbb{P}_D(h(x) \neq y \mid \Delta(\mathcal{H}')) - \mathbb{P}_D(h'(x) \neq y \mid \Delta(\mathcal{H}')),$$

as by definition, $h, h'$ have the same conditional loss on $\Delta(\mathcal{H}')^c$. Inspired by this observation, the idea is to label examples in $\Delta(\mathcal{H}')$, and ignore those outside of it. This allows the learner to learn about the relative performance of classifiers while saving on the labels of examples falling in $\Delta(\mathcal{H}')^c$.

In running a DBAL algorithm, one hopes certain classifiers quickly reveal themselves to be empirically so much worse on $\Delta(\mathcal{H}')$ than the current ERM hypothesis, that by standard concentration bounds, they can be inferred to be worse than $\epsilon$-optimal on $D$ with high probability. Elimination of these classifiers shrinks the disagreement region, allowing the labeling to become further fine-grained. Given the above loss decomposition, this leads to consistent active learning strategies.

## 4.2 Labeling in the Disagreement Region: No Longer Enough

In the multi-group setting, the strategy of comparing performance of classifiers solely on $\Delta(\mathcal{H}')$ breaks down. Although the classifiers in $\mathcal{H}'$ still agree in $\Delta(\mathcal{H}')^c$, this is not enough to infer differences in the worst case error over groups $L_{\mathcal{G}}^{\max}$; this is because differences in performance on $\Delta(\mathcal{H}')$ are not generally representative of differences in absolute errors over group distributions. The following simple example makes this concrete.

**Example 1.** Consider the task of determining which of two classifiers $h$ and $h'$ has lower worst case error over distributions $D_1$ and $D_2$ with marginal supports $S_1 \subseteq \mathcal{X}$ and $S_2 \subseteq \mathcal{X}$. Let their disagreement region be denoted by $\Delta = \{x \in \mathcal{X} : h(x) \neq h'(x)\}$, and let $l(f, i, A)$ denote the conditional loss of classifier $f$ on $S_i \cap A$ under $D_i$. Suppose we only know their conditional losses on $\Delta \cap S_1$ and $\Delta \cap S_2$ under $D_1$ and $D_2$, respectively. We see for $h$ that

$$l(h, i, A) = \begin{cases} 49/100 & i = 1, A = \Delta \cap S_1 \\ 52/100 & i = 2, A = \Delta \cap S_2 \\ ? & i = 1, A = \Delta^c \cap S_1 \\ ? & i = 2, A = \Delta^c \cap S_2 \end{cases}$$

and for $h'$ that

$$l(h', i, A) = \begin{cases} 51/100 & i = 1, A = \Delta \cap S_1 \\ 48/100 & i = 2, A = \Delta \cap S_2 \\ ? & i = 1, A = \Delta^c \cap S_1 \\ ? & i = 2, A = \Delta^c \cap S_2 \end{cases}.$$

Consider ignoring the performance of classifiers in $\Delta^c$, and using as a surrogate for the multi-group objective

$$\max_{i \in \{1,2\}} l(h, i, S_i \cap \Delta).$$

In this case, we would choose $h'$ has the better of the two hypotheses.

Suppose now that $\Delta \cap S_1$ and $\Delta \cap S_2$ have mass $\gamma$ under both $D_1$ and $D_2$, and that $l(h, 1, \Delta^c \cap S_1) = l(h', 1, \Delta^c \cap S_1) = 3/10$. Finally, suppose that $l(h, 2, \Delta^c \cap S_2) = l(h', 2, \Delta^c \cap S_2) = 2/10$. Then under the true multi-group objective, by decomposing the group losses, one can compute that $h$ has a lower worst case error over groups $D_1$ and $D_2$ .

Thus, to utilize the disagreement region in multi-group algorithms, we will need to at least label some samples on $\Delta(\mathcal{H}')^c$ as $\mathcal{H}'$ shrinks. The specification of such a strategy is the content of the next section.

# 5 General Agnostic Multi-Group Learning

## 5.1 An Agnostic Algorithm

The basic idea in Algorithm 1 is similar to classical DBAL approaches for a single distribution. We start with the full hypothesis class $\mathcal{H}$, and look to iteratively eliminate hypotheses from contention as we learn about how to classify on each group through targeted labeling.

Our solution to the problem posed to DBAL above is to keep track of the errors of well-performing hypotheses on the complement of the disagreement region in a way that exploits the agreement property. To do this, we construct a two-part estimate for the loss of a hypothesis on a given group. Denote the set of hypotheses still in contention at iteration $i$ is $\mathcal{H}_i$. Let $R_i = \Delta(\mathcal{H}_i)$ and $S_{R_i,g}$ be a labeled sample from $U(R_i)$ and $S_{R_i^c,g}$ be a labeled sample from $U(R_i^c)$. We can now estimate the loss for some $h \in \mathcal{H}_i$ on group $g$ via

$$L_{S;R_i}(h \mid g) := \mathbb{P}_{D_G}(x \in R_i) \cdot L_{S_{R_i,g}}(h) + \mathbb{P}_{D_G}(x \in R_i^c) \cdot L_{S_{R_i^c,g}}(h_{\mathcal{H}_i}),$$

---

**Algorithm 1** General Agnostic Algorithm

---

1: **procedure** MULTI_GROUP_AGNOSTIC($\mathcal{H}, \epsilon, \delta, \{U_g(\cdot)\}_{g=1}^G, \{O_g(\cdot)\}_{g=1}^G$)
2:     $\mathcal{H}_1 \leftarrow \mathcal{H}, I \leftarrow \lceil \log(1/\epsilon) \rceil$
3:     **for** $i \in [I]$ **do**
4:         $R_i \leftarrow \Delta(\mathcal{H}_i)$
5:         $m_i \leftarrow \max_{g' \in [G]} \mathbb{P}_{D_{g'}}(x \in \Delta(\mathcal{H}_i))$
6:         **for** $g \in [G]$ **do**
7:             $\mathcal{S}'_{R_i,g} \leftarrow 1024 \left( \frac{m_i}{\epsilon 2^{I-i}} \right)^2 \left( 2d \log(64/\epsilon) + \ln(8G \lceil \log(1/\epsilon) \rceil / \delta) \right)$ i.i.d. samples
8:                                                         from $U_g(R_i)$
9:             $\mathcal{S}'_{R_i^c,g} \leftarrow \frac{128 \ln(4G \lceil \log(1/\epsilon) \rceil / \delta)}{(\epsilon 2^{I-i})^2}$ i.i.d. samples from $U_g(R_i^c)$
10:             **if** "None" $\in \mathcal{S}'_{R_i,g}$ **then**                         $\triangleright \mathbb{P}_{D_g}(x \in R_i) = 0$ in this case
11:                 $\mathcal{S}_{R_i,g} \leftarrow \emptyset$
12:             **else**
13:                 $\mathcal{S}_{R_i,g} \leftarrow \{(x, O_g(x)) : x \in \mathcal{S}'_{R_i,g}\}$
14:             **end if**
15:             **if** "None" $\in \mathcal{S}'_{R_i^c,g}$ **then**
16:                 $\mathcal{S}_{R_i^c,g} \leftarrow \emptyset$
17:             **else**
18:                 $\mathcal{S}_{R_i^c,g} \leftarrow \{(x, O_g(x)) : x \in \mathcal{S}'_{R_i^c,g}\}$
19:             **end if**
20:         **end for**
21:         $\hat{h}_i = \arg\min_{h \in \mathcal{H}_i} L_{\mathcal{S};R_i}^{\max}(h)$
22:         $\mathcal{H}_{i+1} \leftarrow \left\{ h \in \mathcal{H}_i : L_{\mathcal{S};R_i}^{\max}(h) \leq L_{\mathcal{S};R_i}^{\max}(\hat{h}_i) + 2^{I-i}\epsilon/4 \right\}$
23:     **end for**
24:     **return** $\hat{h} = \arg\min_{h \in \mathcal{H}_{I+1}} L_{\mathcal{S};R_{I+1}}^{\max}(h)$
25: **end procedure**

---

where $L_S(h) := 1/|\mathcal{S}| \sum_{(x,y) \in \mathcal{S}} \mathbb{1}[h(x) \neq y]$ is a standard empirical loss estimate[1], and $h_{\mathcal{H}_i}$ is an *arbitrarily* chosen hypothesis from $\mathcal{H}_i$ that is used in the loss estimate of every $h \in \mathcal{H}_i$. This leads to an unbiased estimator given that every $h \in \mathcal{H}_i$ labels the sample from this part of space in exactly the same way.

The utility of this estimator is that by choosing an arbitrary representative $h_{\mathcal{H}_i}$, we can estimate the loss of all hypotheses still in contention to precision $O(\epsilon)$ on $R_i^c$ with $\tilde{O}(1/\epsilon^2)$ samples, removing the usual dependence of the VC-dimension. On the other hand, as the disagreement region shrinks, $\mathbb{P}_{D_G}(x \in R_i)$ shrinks as well, so while we will still need to invoke uniform convergence to get reliable loss estimates in $R_i$, the precision to which we need to estimate losses in this part of space decreases with every iteration, and eventually the overall dependence on the VC-dimension is diminished. This later observation is the standard source of gains in DBAL [18, 9, 24].

After forming these loss estimates on each group, we construct unbiased loss estimates for the worst case over groups via

$$L_{\mathcal{S};R_i}^{\max}(h) := \max_{g \in \mathcal{G}} L_{\mathcal{S};R_i}(h \mid g).$$

These loss estimates inherit concentration properties from the two-part estimator above. We draw enough samples at each iteration $i$ such that we essentially learn the multi-group problem to precision $2^{\lceil \log(1/\epsilon) \rceil - i}\epsilon$.

We note that Algorithm 1 assumes access to the underlying group marginals measures $\mathbb{P}_{D_G}$. This is common in the active learning literature [18, 20]. Probabilities of events in instance space can be estimated to arbitrary accuracy using only unlabeled data, so this assumption is not dangerous to our goal of lowering label complexities.

---

[1] taken to be an arbitrary constant if $\mathcal{S} = \emptyset$; see the Appendix for details.

## 5.2 Guarantees

Vitally, the scheme given in Algorithm 1 is consistent. It is a lemma of ours that the number of samples drawn at each iteration is sufficiently large that the true error of any $h \in \mathcal{H}_{i+1}$ is no more than $2^{\lceil \log(1/\epsilon) \rceil - i} \epsilon$. Thus, after $\lceil \log(1/\epsilon) \rceil$ iterations, the ERM hypothesis on $L_{\mathcal{S};R_i}^{\max}(\cdot)$ is then $\epsilon$-optimal with high probability.

We can bound the label complexity of the algorithm using standard techniques from DBAL. A ubiquitous quantity in the analysis of disagreement-based schemes is that of the "disagreement coefficient" [9, 25]. The general idea is that the disagreement coefficient bounds the rate of decrease in $r$ of the measure of the the disagreement region of a ball of radius $r$ around $h^*$ in the pseudo-metric $\rho_g(h, h') := \mathbb{P}_{D_g}(h(x) \neq h'(x))$. Precisely, we use the following definition of the disagreement coefficient in our analysis [10, 24]: given a group $D_g$, the disagreement coefficient on $g$ is

$$\theta_g := \sup_{h \in \mathcal{H}} \sup_{r' \geq 2\nu + \epsilon} \frac{\mathbb{P}_{D_g}(x \in \Delta(B_g(h, r')))}{r'},$$

where $B_g(h, r') := \{h' \in \mathcal{H} : \rho_g(h, h') \leq r'\}$ is a ball of radius $r'$ about $h$ in pseudo-metric $\rho_g$. We further notate the maximum disagreement coefficient over the groups $\mathcal{G}$ as $\theta_{\mathcal{G}} := \max_g \theta_g$.

The disagreement coefficient is trivially bounded above by $1/\epsilon$, but can be bounded independently of $\epsilon$ in many cases [10, 25]. For example, when $\mathcal{H}$ is the class of linear separators in $d$ dimensions and the underlying marginal distribution is uniform over the Euclidean unit sphere, the disagreement coefficient is $\Theta(\sqrt{d})$ [9].

**Theorem 1.** *For all $\epsilon > 0$, $\delta \in (0, 1)$, collections of groups $\mathcal{G}$, and hypothesis classes $\mathcal{H}$ with $d < \infty$, with probability $\geq 1 - \delta$, the output $\hat{h}$ of Algorithm 1 satisfies*

$$L_{\mathcal{G}}^{\max}(\hat{h}) \leq L_{\mathcal{G}}^{\max}(h^*) + \epsilon,$$

*and its label complexity is bounded by*

$$\tilde{O}\left(G \, \theta_{\mathcal{G}}^2 \left(\frac{\nu^2}{\epsilon^2} + 1\right) \left(d \log(1/\epsilon) + \log(1/\delta)\right) \log(1/\epsilon) + \frac{G \log(1/\delta) \log(1/\epsilon)}{\epsilon^2}\right).$$

Here, the $\tilde{O}$ notation hides factors of $\log(\log(1/\epsilon))$ and $\log(G)$; we leave all proofs for the Appendix.

Theorem 1 tell us that Algorithm 1 enjoys the following upside over passive and collaborative learning approaches: the dependence on the standard interaction of the VC-dimension $d$ and $1/\epsilon^2$ is removed, and replaced with $G d \theta_{\mathcal{G}}^2 \log^2(1/\epsilon) \nu^2/\epsilon^2$, which in settings with small disagreement coefficients and low noise rates, will be significant for small $\epsilon$.

## 5.3 Comparison to Lower Bounds in Collaborative Learning

We compare our label complexity guarantees to results in collaborative learning, where the learner has the power to ask for samples from specific group distributions, but not selectively label these samples. This is a strictly more demanding comparison than to pure passive settings, but a fair one, given that active learners have the option of executing any collaborative learning strategy. In collaborative learning, for finite hypothesis classes $\mathcal{H}$, it is known that

$$\Omega\left(\frac{\log(|\mathcal{H}|)}{\epsilon^2} + \frac{G \log(\min(|\mathcal{H}|, G)/\delta)}{\epsilon^2}\right)$$

total labels over all groups are necessary [3]. We consider comparing this lower bound to a simplified version of the label complexity guarantee in Theorem 1:

$$\tilde{O}\left(dG\theta_{\mathcal{G}}^2 \log^2(1/\epsilon) + \frac{G \log(1/\epsilon)}{\epsilon^2}\right),$$

thus implicitly assuming $\nu$ is neglectably small, and making all comparisons up to factors logarithmic in $G$, $1/\delta$ and $\log(1/\epsilon)$. This former assumption is a standard assumption under which we may hope an agnostic active learner to succeed [15].

---

**Algorithm 2** Group Realizable Algorithm

---

    **procedure** GROUP_REALIZABLE($\mathcal{H}, \epsilon, \delta$, active learner $\mathcal{A}$, $\{U_g(\cdot)\}_{g=1}^G$, $\{O_g(\cdot)\}_{g=1}^G$))

        **for** $g \in [G]$ **do**

            $\hat{h}_g \leftarrow \mathcal{A}(\mathcal{H}, \epsilon/6, \delta/2G, U_g(\mathcal{X}), O_g)$

            $S'_g \leftarrow 144/\epsilon^2 \left( 2d \ln(24/\epsilon) + \ln(8G/\delta) \right)$ samples from oracle $U_g(\mathcal{X})$

            $\hat{S}_g \leftarrow \left\{ \left( x, \hat{h}_g(x) \right) : x \in S'_g \right\}$

        **end for**

        **return** $\hat{h} = \arg\min_{h \in \mathcal{H}} \max_{g \in [G]} \frac{1}{|\hat{S}_g|} \sum_{(x,\hat{y}) \in \hat{S}_g} \mathbb{1}\left[ h(x) \neq \hat{y} \right]$

    **end procedure**

---

Even the simplified upper bound does not admit the cleanest comparison this to lower bound, due to our excess factor of $\log(1/\epsilon)$ in the second term. However, it does showcase that while we pay slightly more per group than necessary, under conditions amenable to active learning, we pay significantly less per dimension $d$. Particularly for small $\epsilon$, one can see that's approximately sufficient that $G < o(\theta_{\mathcal{G}}^2 (\log(1/\epsilon)\epsilon)^2)^{-1})$ for the simplified upper bound to beat the lower bound.

For a more fine-grained comparison that in some sense underestimates the power of Algorithm 1, assume that the following condition governs the relationship of $G$, $d$, and $\epsilon$:

$$G \log(1/\epsilon) \leq d < \left( \theta_{\mathcal{G}}^2 \epsilon^2 \log^2(1/\epsilon) \right)^{-1}.$$

Then the simplified bound is smaller in order than the lower bound above.

## 6 Group Realizable Learning

A special case of the learning problem, where extreme active learning gains can be readily seen, comes when the hypothesis class $\mathcal{H}$ achieves zero noise rate on each group $D_g$. This setting has been considered in the passive "multi-group learning" literature [6]. Formally speaking, in the group realizable setting, the following condition holds:

$$\forall g \in [G], \exists h_g^* \in \mathcal{H} \ s.t. \ L_{\mathcal{G}}(h_g^* \mid g) = 0,$$

i.e. for all groups in the collection $\mathcal{G}$, there is some hypothesis achieving 0 error on that group. Note that this differs from the fully realizable setting where there is some $h^* \in \mathcal{H}$ with $L_{\mathcal{G}}^{\max}(h^*) = 0$. While fully realizable implies group realizable, the converse is not true. Thus, group realizability represents an intermediate regime between the realizable setting and the full agnostic settings.

### 6.1 Algorithm

In the group realizable case, it is possible to show a reduction of the problem of active learning over hypothesis classes with respect to a single distribution.

This can be accomplished as follows. For each $D_g$, we call as a subroutine an active learner that is guaranteed to find an order $\epsilon$-optimal hypothesis $\hat{h}_g^*$ with high probability over it's queries. It then gathers new unlabeled samples from each $D_g$, and instead of requesting labels from $O_g(\cdot)$, labels each unlabeled point with $\hat{h}_g^*$. The final step is to do an empirical risk minimization on these artificially labeled samples with respect to the multi-group objective. See Algorithm 2 for a formal specification of the strategy.

### 6.2 Guarantees

The strategy given in Algorithm 2 leads to a consistent active learning scheme, provided the active learners called as subroutines have standard guarantees that can be inherited.

Theorem 2 gives a guarantee to this end. The proof follows from an argument similar to one used in [10] - because the subroutine calls return hypotheses with near 0 error on each group, the artificially labeled training set used in the ERM step looks nearly identical to a counterfactual training set for the ERM step constructed by querying labels $O_g(x)$ for each unlabeled $x$. This is similar to the idea in [24]. We present Theorem 2 assuming access to a classical, realizable active learner due to [26].

**Theorem 2.** *Suppose Algorithm 2 is run with the active learner $\mathcal{A}_{CAL}$ of [26]. Then for all $\epsilon > 0$, $\delta \in (0,1)$, hypothesis classes $\mathcal{H}$ with $d < \infty$, and collections of groups $\mathcal{G}$ with the group realizability property under $\mathcal{H}$, with probability $\geq 1 - \delta$, the output $\hat{h}$ satisfies*

$$L_{\mathcal{G}}^{\max}(\hat{h}) \leq L_{\mathcal{G}}^{\max}(h^*) + \epsilon,$$

*and the number of labels requested is*

$$\tilde{O}\bigg( dG\theta_{\mathcal{G}} \log(1/\epsilon) \bigg).$$

Thus, when disagreement coefficients across the collection of groups are bounded independently of $\epsilon$, the usual, passive dependence on $1/\epsilon^2$ is replaced by $\log(1/\epsilon)$.

In the passive multi-group setting of [7], it has been shown that $\tilde{O}\left(\log(|\mathcal{H}|)/\gamma\epsilon\right)$ samples are sufficient for group realizable learning, where we recall $\gamma$ is a lower bound on the probability of getting a sample in each group [6].

## 7 Full Agnostic Approximation

### 7.1 Inconsistency of the Reduction in the Full Agnostic Regime

Algorithm 2 admits clean analysis, and nicely harnesses the power of realizable active learners for a single distribution. One might wonder if a similar strategy might provide a consistent strategy in full agnostic regime. Unfortunately, the direct application of Algorithm 2 using agnostic learners does not yield a consistent active learning algorithm. In fact, consistency fails even when for each $g \in [G]$, $h_g^*$ is the Bayes optimal classifier on $D_g$, and $\nu_g := \inf_{h \in \mathcal{H}} L_{\mathcal{G}}(h \mid g)$ is small. This lack of consistency comes down to the fact that labeling with the Bayes optimal underestimates noise rates on each group, which in turn may bias the output of the ERM step.

### 7.2 A $3\nu$-Approximation Algorithm

Although the strategy of creating an artificially labeled training set with near-optimal hypotheses on each group fails outside of the group realizable case, it possesses a nice approximation property.

We give a guarantee to this end in Theorem 3. It states that if we call an active learner with agnostic guarantees on each group $D_g$, and then use the outputs $\hat{h}_g^*$ to artificially label a new batch of unlabeled data from each group, using ERM on this artificially labeled data gives at worst a $2\nu + \epsilon$-optimal hypothesis with high probability.

**Theorem 3.** *Suppose Algorithm 2 is run with the agnostic active learner $\mathcal{A}_{DHM}$ of [15]. Then for all $\epsilon > 0$, $\delta \in (0,1)$, hypothesis classes $\mathcal{H}$ with $d < \infty$, and collections of groups $\mathcal{G}$, with probability $\geq 1 - \delta$, the output $\hat{h}$ satisfies*

$$L_{\mathcal{G}}^{\max}(\hat{h}) \leq L_{\mathcal{G}}^{\max}(h^*) + 2 \cdot \max_{g \in [G]} \nu_g + \epsilon \leq 3 \cdot L_{\mathcal{G}}^{\max}(h^*) + \epsilon,$$

*and the number of labels requested is*

$$\tilde{O}\left( dG\theta_{\mathcal{G}}\left( \log^2(1/\epsilon) + \frac{\nu^2}{\epsilon^2} \right) \right).$$

The proof is very similar to that of Theorem 2, but notes in addition that $\hat{h}_g^*$ mislabels on a roughly $\nu_g$-fraction of the unlabeled samples from each group $G$. This allows us to upper bound the distortion of the ERM step.

## 8 Conclusion

In this work, we have taken a first look at active multi-group learning. Though the design of general agnostic strategies in this setting is quite challenging, an interesting future direction may be the search for strategies that work in more specific cases, for exampling extending our work in the group realizable setting. In particular, the search for algorithms with small label complexities under specific low-noise conditions, such as Tsybakov noise on each $D_g$, may prove fruitful [27].

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

# 9 Appendix

## 9.1 Guarantees for General Agnostic Algorithm

In this section, we give proofs for the guarantees of Algorithm 1. We begin with some definitions, starting with how empirical loss estimates are made.

**Definition 1.** *Given a hypothesis $h \in \mathcal{H}$, and a set of pairs $\mathcal{S} = \{(x_i, y_i) : x_i \in \mathcal{X}, y_i \in \mathcal{Y}\}_{i=1}^{N}$, let*

$$L_{\mathcal{S}}(h) := \frac{1}{N} \left( \sum_{i=1}^{N} \mathbb{1}[h(x_i) \neq y_i] \right)$$

*the standard empirical loss of $h$ on $\mathcal{S}$. Let $L_{\emptyset}(h) := 1$.*

The convention to let $L_{\emptyset}(h) = 1$ allows us to "collapse" the two-part loss estimates in the case the probability of drawing an unlabeled sample in a specific region is 0; under the specification of the algorithm, $\mathcal{S} = \emptyset$ if and only if the probability of a sample falling in the disagreement region or its complement is 0 under $D_g$, in which case we can safely ignore estimation in one of these regions.

**Definition 2.** *Given a set of classifiers $\mathcal{H}' \subseteq \mathcal{H}$, we say "$\mathcal{H}'$ agrees on a subset $S \subseteq \mathcal{X}$" if for each $x \in S$ and for each pair $(h, h') \in \mathcal{H}' \times \mathcal{H}'$, it holds that $h(x) = h'(x)$.*

We now recall the two-part estimator for the loss of a hypothesis introduced above.

**Definition 3.** *Fix a group distribution $D_g$, some $\mathcal{H}' \subseteq \mathcal{H}$, a hypothesis $h \in \mathcal{H}'$, and some $R \subseteq \mathcal{X}$ which is measurable with respect to each marginal of $D_g$ and for which $\mathcal{H}'$ agrees on $R^c$. Given sets of pairs $\mathcal{S}_{R,g}$ and $\mathcal{S}_{R^c,g}$, and some arbitrarily chosen classifier $h_{\mathcal{H}'} \in \mathcal{H}'$, let*

$$L_{\mathcal{S};R}(h \mid g) := \mathbb{P}_{D_g}(x \in R) \cdot L_{\mathcal{S}_{R,g}}(h) + \mathbb{P}_{D_g}(x \in R^c) \cdot L_{\mathcal{S}_{R^c,g}}(h_{\mathcal{H}'}).$$

As mentioned in the main body, $h_{\mathcal{H}'}$ must be used in the estimate of the loss under $D_g$ in the "agreement region" for all $h \in \mathcal{H}$. The extent to which this estimator is useful can be captured by standard uniform convergence arguments. To this end, we first introduce a function that will prove to control its deviations nicely.

**Definition 4.** *Given a confidence parameter $\delta \in (0, 1)$, a group distribution $D_g \in \mathcal{G}$, some $R \subseteq \mathcal{X}$ that is measurable with respect to each marginal $D_g \in \mathcal{G}$, and sample sizes $m, m' > 0$, define the function*

$$\Gamma_g(\delta, R, m, m') := \begin{cases} \mathbb{P}_{D_g}(x \in R)\left( \frac{1}{m} + \sqrt{\frac{\ln(8/\delta) + d\ln(2em/d)}{m}} \right) + \sqrt{\frac{\ln(4/\delta)}{2m'}} \\ \qquad\qquad\qquad \text{if } \mathbb{P}_{D_g}(x \in R) > 0, \mathbb{P}_{D_g}(x \in R^c) > 0 \\[2mm] \frac{1}{m} + \sqrt{\frac{\ln(8/\delta) + d\ln(2em/d)}{m}} \\ \qquad\qquad\qquad \text{if } \mathbb{P}_{D_g}(x \in R) > 0, \mathbb{P}_{D_g}(x \in R^c) = 0 \\[2mm] \sqrt{\frac{\ln(4/\delta)}{2m'}} \\ \qquad\qquad\qquad \text{if } \mathbb{P}_{D_g}(x \in R) = 0, \mathbb{P}_{D_g}(x \in R^c) > 0. \end{cases}$$

**Lemma 1.** *Fix $\delta \in (0, 1)$, a set of group distributions $\mathcal{G}$, and a group distribution $D_g \in \mathcal{G}$ arbitrarily. Further, fix a subset $R \subseteq \mathcal{X}$ measurable with respect to each marginal of $D_g \in \mathcal{G}$, and a set of classifiers $\mathcal{H}' \subseteq \mathcal{H}$ with the property that $\mathcal{H}'$ agree on $R^c$. Suppose we query $m > 0$ unlabeled samples from $U_g(R)$, and $m' > 0$ samples from $U_g(R^c)$. Suppose further that we label the output via calls to $O_g(\cdot)$, forming the labeled samples $\mathcal{S}_{R,g}$ and $\mathcal{S}_{R^c,g}$, respectively; if either $\mathbb{P}_{D_g}(x \in R) = 0$ or $\mathbb{P}_{D_g}(x \in R^c)$, then we set the corresponding sample to be $\emptyset$. Then with probability $\geq 1 - \delta$, it holds for all $h \in \mathcal{H}'$ that*

$$|L_{\mathcal{G}}(h \mid g) - L_{\mathcal{S};R}(h \mid g)| \leq \Gamma_g(\delta, R, m, m').$$

*Further, for all $\gamma > 0$, if $m \geq \frac{16(\mathbb{P}_{D_g}(x \in R))^2}{\gamma^2}(2d\ln(8/\gamma) + \ln(8/\delta))$ and $m' \geq \frac{2\ln(4/\delta)}{\gamma^2}$, then $\Gamma_g(\delta, R, m, m') < \gamma$.*

*Proof.* We begin with the case where both $\mathbb{P}_{D_g}(x \in R) \neq 0$ and $\mathbb{P}_{D_g}(x \in R^c) \neq 0$. In this case, we are able to draw unlabeled samples from both regions, and neither $\mathcal{S}_{R,g}$ nor $\mathcal{S}_{R,g}$ is $\emptyset$.

By a lemma of Vapnik [28], we have that with probability $\geq 1 - \delta/2$ over the draw of $m$ samples from $U_g(R)$ and their labeling via $O_g(\cdot)$, that simultaneously for each $h \in \mathcal{H}'$:

$$\left| L_{\mathcal{S}_{R,g}}(h) - \mathbb{P}_{D_g}(h(x) \neq y| \, x \in R) \right| \leq \frac{1}{m} + \sqrt{\frac{\ln(8/\delta) + d\ln(2em/d)}{m}}.$$

In $R^c$, all $h \in \mathcal{H}'$ agree, and so estimating the conditional loss for each $h \in \mathcal{H}'$ in this region is as statistically hard as estimating a single Bernoulli parameter, which we do by arbitrarily choosing a classifier to use for the loss estimate in this part of space. Thus, by definition of the two-part estimator and Hoeffding's inequality [29], we have with probability $\geq 1 - \delta/2$ for all $h \in \mathcal{H}'$ simultaneously

$$\left| L_{\mathcal{S}_{R^c,g}}(h_{\mathcal{H}'}) - \mathbb{P}_{D_g}(h(x) \neq y| \, x \in R^c) \right| \leq \sqrt{\frac{\ln(4/\delta)}{2m'}}.$$

By a union bound, with probability $\geq 1 - \delta$, both of these events take place, and so for all $h \in \mathcal{H}'$ simultaneously,

$$\begin{aligned} L_{\mathcal{G}}(h \mid g) &= \mathbb{P}_{D_g}\big(h(x) \neq y \mid x \in R\big) \cdot \mathbb{P}_{D_g}(x \in R) \\ &\quad + \mathbb{P}_{D_g}(h(x) \neq y \mid x \in R^c) \cdot \mathbb{P}_{D_g}(x \in R^c) \\ &\leq \left( L_{\mathcal{S}_{R,g}}(h) + 1/m + \sqrt{(\ln(8/\delta) + d\ln(2em/d))/m} \right) \cdot \mathbb{P}_{D_g}(x \in R) \\ &\quad + \left( L_{\mathcal{S}_{R^c,g}}(h_{\mathcal{H}'}) + \sqrt{\ln(4/\delta)/2m'} \right) \cdot \mathbb{P}_{D_g}(x \in R^c) \\ &\leq L_{\mathcal{S};R}(h \mid g) + \Gamma_g(\delta, R, m, m'). \end{aligned}$$

The lower bound leading to the absolute value is analogous. Vapnik [28] also tells us that for any $\gamma' > 0$, a sample of size $m \geq \frac{4}{\gamma'^2}\left(2d\ln(4/\gamma') + \ln(8/\delta)\right)$ is sufficient to yield

$$1/m + \sqrt{(\ln(8/\delta) + d\ln(2em/d))/m} < \gamma'.$$

Let $\gamma' = \gamma/2\mathbb{P}_{D_g}(x \in R)$. Thus, substituting for $\gamma'$ and bounding the probability inside the natural log above by 1,

$$m \geq \mathbb{P}_{D_g}(x \in R)^2 \frac{16}{\gamma^2}\left(2d\ln(8/\gamma) + \ln(8/\delta)\right)$$

implies that

$$\frac{1}{m} + \sqrt{\frac{\ln(8/\delta) + d\ln(2em/d)}{m}} < \frac{\gamma}{2\mathbb{P}_{D_g}(x \in R)}.$$

As a corollary to Hoeffding, if $m' \geq 2\ln(4/\delta)/\gamma^2$, then $\sqrt{\log(4/\delta)/2m'} < \gamma/2$. Thus, we may write

$$\Gamma_g(\delta, R, m, m') = \mathbb{P}_{D_g}(x \in R)\left(\frac{1}{m} + \sqrt{\frac{\ln(8/\delta) + d\ln(2em/d)}{m}}\right) + \sqrt{\frac{\ln(4/\delta)}{2m'}} < \gamma/2 + \gamma/2 = \gamma.$$

Now suppose that $\mathbb{P}_{D_g}(x \in R^c) = 0$. In this case, we have $\mathcal{S}_{R^c,g} = \emptyset$. Again, we have that with probability $\geq 1 - \delta/2$,

$$\left| L_{\mathcal{S}_{R,g}}(h) - \mathbb{P}_{D_g}(h(x) \neq y| \, x \in R) \right| \leq \frac{1}{m} + \sqrt{\frac{\ln(8/\delta) + d\ln(2em/d)}{m}}.$$

When $\mathbb{P}_{D_g}(x \in R^c) = 0$, it holds that $\mathbb{P}_{D_g}(x \in R) = 1$, and so

$$\begin{aligned} L_{\mathcal{G}}(h \mid g) &= \mathbb{P}_{D_g}(h(x) \neq y \mid x \in R) \cdot \mathbb{P}_{D_g}(x \in R) \\ &\quad + \mathbb{P}_{D_g}(h(x) \neq y \mid x \in R^c) \cdot \mathbb{P}_{D_g}(x \in R^c) \\ &= \mathbb{P}_{D_g}(h(x) \neq y \mid x \in R) \\ &\leq L_{\mathcal{S}_{R,g}}(h) + 1/m + \sqrt{(\ln(8/\delta) + d\ln(2em/d))/m} \\ &= L_{\mathcal{S};R}(h \mid g) + \Gamma_g(\delta, R, m, m'), \end{aligned}$$

where the final equality comes from fact that $\mathbb{P}_{D_g}(x \in R^c) = 0$ and $\mathbb{P}_{D_g}(x \in R) = 1$, as well as the definitions of $L_{\mathcal{S};R}(h \mid g)$ and $\Gamma_g(\delta, R, m, m')$. Similarly to the above, if we let $\gamma' = \gamma/2\mathbb{P}_{D_g}(x \in R) = \gamma/2$, then

$$m \geq \frac{16}{\gamma^2} \left(2d\ln(8/\gamma) + \ln(8/\delta)\right)$$

implies that

$$\frac{1}{m} + \sqrt{\frac{\ln(8/\delta) + d\ln(2em/d)}{m}} < \frac{\gamma}{2},$$

which by the definition of $\Gamma_g(\delta, R, m, m')$ when $\mathbb{P}_{D_g}(x \in R^c) = 0$ gives us $\Gamma_g(\delta, R, m, m') < \gamma/2 < \gamma$. The case where $\mathbb{P}_{D_g}(x \in R) = 0$ follows the previous argument for when $\mathbb{P}_{D_g}(x \in R^c) = 0$. $\qquad\square$

**Definition 5.** *Given a collection of group distributions $\mathcal{G}$, some $\mathcal{H}' \subseteq \mathcal{H}$, a hypothesis $h \in \mathcal{H}'$, some subset $R \subseteq \mathcal{X}$ measurable with respect to each marginal of $D_g \in \mathcal{G}$, and labeled samples $\mathcal{S}_{R,k}$ and $\mathcal{S}_{R^c,k}$, we define the empirical estimate of the multi-group loss of $h$ parameterized by $R$ via*

$$L_{\mathcal{S};R}^{\max}(h) := \max_{g \in [G]} L_{\mathcal{S};R}(h \mid g).$$

Having recalled the way in which we form empirical estimates for the group worst-case loss of a given hypothesis, we can show a simple concentration lemma for this group worst-case loss estimator using the concentration property for individual groups proved in Lemma 1,

**Lemma 2.** *Fix $\delta \in (0, 1)$, a set of group distributions $\mathcal{G}$, a subset $R \subseteq \mathcal{X}$ measurable with respect to each marginal of $D_g \in \mathcal{G}$, and a set of classifiers $\mathcal{H}' \subseteq \mathcal{H}$ that agree on $R^c$. Suppose for each $g \in [G]$, we query $m_g > 0$ unlabeled samples from $U_g(R)$, and $m'_g > 0$ samples from $U_g(R^c)$. Suppose further that we label the outputs via calls to $O_g(\cdot)$, forming the labeled samples $\mathcal{S}_{R,g}$ and $\mathcal{S}_{R^c,g}$, respectively, for each $g \in [G]$; if $\mathbb{P}_{D_g}(x \in R) = 0$ or $\mathbb{P}_{D_g}(x \in R^c) = 0$, then we set the corresponding sample to be $\emptyset$. Then with probability $\geq 1 - \delta$, it holds for all $h \in \mathcal{H}'$ that*

$$\left| L_{\mathcal{G}}^{\max}(h) - L_{\mathcal{S};R}^{\max}(h) \right| \leq \max_{g' \in [G]} \Gamma_{g'}(\delta/G, m_{g'}, m'_{g'}).$$

*Proof.* By Lemma 1 and a union bound, it holds with probability $\geq 1 - \delta$ that on all $D_g$, for all $h \in \mathcal{H}'$ simultaneously, that

$$|L_{\mathcal{G}}(h \mid g) - L_{\mathcal{S};R}(h \mid g)| \leq \Gamma_g(\delta/G, m_g, m'_g).$$

Thus we may write

$$
\begin{aligned}
\left| L_{\mathcal{G}}^{\max}(h) - L_{\mathcal{S};R}^{\max}(h) \right| &= \left| \max_{g' \in [G]} L_{\mathcal{G}}(h \mid g') - \max_{g' \in [G]} L_{\mathcal{S};R}(h \mid g) \right| \\
&\leq \max_{g' \in [G]} \left| L_{\mathcal{G}}(h \mid g') - L_{\mathcal{S};R}(h \mid g') \right| \\
&\leq \max_{g' \in [G]} \Gamma_{g'}(\delta/G, m_{g'}, m'_{g'}).
\end{aligned}
$$

$\qquad\square$

We now use Lemma 2 to show that Algorithm 1 is conservative enough that the optimal hypothesis $h^*$ is never eliminated from contention throughout the run of the algorithm with high probability.

**Lemma 3.** *Fix $\delta \in (0, 1)$, a collection of group distributions $\mathcal{G}$, and a hypothesis class $\mathcal{H}$ with $d < \infty$ arbitrarily. With probability $\geq 1 - \delta$, it holds after each iteration $i$ of Algorithm 1 that $h^* \in \mathcal{H}_{i+1}$.*

*Proof.* By Lemmas 1 and 2, and a union bound over iterations, the number of samples labeled at each iteration is sufficient for us to conclude that with probability $\geq 1 - \delta$, for for every iteration $i$

and for each $h \in \mathcal{H}_i$, it holds that[2]

$$|L_{\mathcal{S};R_i}^{\max}(h) - L_{\mathcal{G}}^{\max}(h)| \leq 2^{I-i}\epsilon/8.$$

We give an inductive argument conditioned on this high probability event. When $i = 1$, we have $h^* \in \mathcal{H}_1$ because $\mathcal{H}_1 = \mathcal{H}$, and $h^* \in \mathcal{H}$ by definition. If $h^* \in \mathcal{H}_i$ for $i \geq 1$, then $h^* \in \mathcal{H}_{i+1}$ if and only if

$$L_{\mathcal{S};R_i}^{\max}(h^*) \leq L_{\mathcal{S};R_i}^{\max}(\hat{h}_i) + 2^{I-i}\epsilon/4.$$

When for each $h \in \mathcal{H}_i$, it holds that $|L_{\mathcal{S};R_i}^{\max}(h) - L_{\mathcal{G}}^{\max}(h)| \leq 2^{I-i}\epsilon/8$, we may write

$$
\begin{aligned}
L_{\mathcal{S};R_i}^{\max}(h^*) - L_{\mathcal{S};R_i}^{\max}(\hat{h}_i) &\leq L_{\mathcal{S};R_i}^{\max}(h^*) - L_{\mathcal{G}}^{\max}(h^*) + L_{\mathcal{G}}^{\max}(\hat{h}_i) - L_{\mathcal{S};R_i}^{\max}(\hat{h}_i) \\
&\leq \left| L_{\mathcal{S};R_i}^{\max}(h^*) - L_{\mathcal{G}}^{\max}(h^*) \right| + \left| L_{\mathcal{G}}^{\max}(\hat{h}_i) - L_{\mathcal{S};R_i}^{\max}(\hat{h}_i) \right| \\
&\leq 2^{I-i}\epsilon/8 + 2^{I-i}\epsilon/8 \\
&= 2^{I-i}\epsilon/4,
\end{aligned}
$$

where the first inequality comes from the optimality of $h^*$. Thus, we must have $h \in \mathcal{H}_{i+1}$. $\qquad\square$

Now, using the fact that the optimal hypothesis stays in contention throughout the run of the algorithm, we can give a guarantee on the true error of each hypothesis $h \in \mathcal{H}_{i+1}$. The idea is that using concentration and the small empirical error of each $h \in \mathcal{H}_{i+1}$, we can say that the true errors of each $h \in \mathcal{H}_{i+1}$ are similar to the true errors of the ERM hypothesis $\hat{h}_i$, and then use the true error of $\hat{h}_i$ as a reference point to which we can compare the true error of $h \in \mathcal{H}_{i+1}$ and $h^*$.

**Lemma 4.** *Fix $\delta \in (0,1)$, a collection of group distributions $\mathcal{G}$, and a hypothesis class $\mathcal{H}$ with $d < \infty$ arbitrarily. Then with probability $\geq 1 - \delta$, after every iteration $i$ of Algorithm 1, it holds for all $h \in \mathcal{H}_{i+1}$ that*

$$\left| L_{\mathcal{G}}^{\max}(h) - L_{\mathcal{G}}^{\max}(h^*) \right| \leq 2^{I-i}\epsilon.$$

*Proof.* If $h \in \mathcal{H}_{i+1}$, then by the specification of the algorithm, it holds that

$$L_{\mathcal{S};R_i}^{\max}(h) - L_{\mathcal{S};R_i}^{\max}(\hat{h}_i) \leq 2^{I-i}\epsilon/4.$$

Because $\hat{h}_i$ is the ERM hypothesis at iteration $i$, it holds that $L_{\mathcal{S};R_i}^{\max}(\hat{h}_i) - L_{\mathcal{S};R_i}^{\max}(h) \leq 0 < 2^{I-i}\epsilon/4$, and thus we may conclude

$$\left| L_{\mathcal{S};R_i}^{\max}(h) - L_{\mathcal{S};R_i}^{\max}(\hat{h}_i) \right| \leq 2^{I-i}\epsilon/4.$$

By Lemma 2 and the number of samples labeled at each iteration, with probability $\geq 1 - \delta$, it holds for all iterations and for all $h \in \mathcal{H}_i$ that

$$\left| L_{\mathcal{S};R_i}^{\max}(h) - L_{\mathcal{G}}^{\max}(h) \right| \leq 2^{I-i}\epsilon/8.$$

Conditioned on this event, if $h \in \mathcal{H}_{i+1}$, we have

$$
\begin{aligned}
\left| L_{\mathcal{G}}^{\max}(h) - L_{\mathcal{G}}^{\max}(\hat{h}_i) \right| &= \left| L_{\mathcal{G}}^{\max}(h) - L_{\mathcal{S};R_i}^{\max}(h) + L_{\mathcal{S};R_i}^{\max}(h) - L_{\mathcal{S};R_i}^{\max}(\hat{h}_i) + L_{\mathcal{S};R_i}^{\max}(\hat{h}_i) - L_{\mathcal{G}}^{\max}(\hat{h}_i) \right| \\
&\leq \left| L_{\mathcal{G}}^{\max}(h) - L_{\mathcal{S};R_i}^{\max}(h) \right| + \left| L_{\mathcal{S};R_i}^{\max}(h) - L_{\mathcal{S};R_i}^{\max}(\hat{h}_i) \right| + \left| L_{\mathcal{S};R_i}^{\max}(\hat{h}_i) - L_{\mathcal{G}}^{\max}(\hat{h}_i) \right| \\
&\leq 2^{I-i}\epsilon/8 + 2^{I-i}\epsilon/4 + 2^{I-i}\epsilon/8 \\
&= 2^{I-i}\epsilon/2.
\end{aligned}
$$

By Lemma 3, it holds that $h^* \in \mathcal{H}_{i+1}$ whenever $\left| L_{\mathcal{S};R_i}^{\max}(h) - L_{\mathcal{G}}^{\max}(h) \right| \leq 2^{I-i}\epsilon/8$ for all $h \in \mathcal{H}_i$ at all iterations. Thus, this bound on the true error difference with the ERM $\hat{h}_i$ applies to $h^*$, and we may write for arbitrary $h \in \mathcal{H}_{i+1}$ that

$$\left| L_{\mathcal{G}}^{\max}(h) - L_{\mathcal{G}}^{\max}(h^*) \right| \leq \left| L_{\mathcal{G}}^{\max}(h) - L_{\mathcal{G}}^{\max}(\hat{h}_i) \right| + \left| L_{\mathcal{G}}^{\max}(\hat{h}_i) - L_{\mathcal{G}}^{\max}(h^*) \right| \leq 2^{I-i}\epsilon,$$

which is the desired result. $\qquad\square$

---

[2] We do not directly apply Lemma 1 with $\gamma = \epsilon 2^{I-i}/8$ here. We use this quantity in the outer dependence on $\gamma$ of Lemma 1, but for the natural log dependence on $\gamma$, we sub in $\epsilon/8$ to simplify the analysis. Thus we take slightly more samples than Lemma 1 directly suggests. Because we take the largest probability of the disagreement region over groups as $m_i$, it holds that $m_g$ is at the smallest the sample size suggested by Lemma 1 for each $g$.

**Definition 6.** *Given a group distribution $D_g \in \mathcal{G}$, a hypothesis $h \in \mathcal{H}$, and a radius $r \geq 0$, let the "$D_g$ - disagreement ball in $\mathcal{H}$ of radius $r$ about $h$" be*

$$B_g(h, r) := \left\{ h' \in \mathcal{H} : \rho_g(h, h') \leq r \right\},$$

*where $\rho_g(h, h') := \mathbb{P}_{D_g}(h(x) \neq h'(x))$.*

**Definition 7.** *Given a group distribution $D_g \in \mathcal{G}$ and a hypothesis class $\mathcal{H}$, let the "disagreement coefficient" of $D_g$ be defined as*

$$\theta_g := \sup_{h \in \mathcal{H}} \sup_{r' \geq 2\nu + \epsilon} \frac{\mathbb{P}_{D_g}(x \in \Delta(B_g(h, r')))}{r'}.$$

*We further define the disagreement coefficient over a collection of group distributions $\mathcal{G}$ as*

$$\theta_{\mathcal{G}} := \max_{g' \in [G]} \theta_{g'}.$$

Given these definitions, we are now ready to state the main theorem. The consistency comes from what we showed in Lemma 4: as the true error for each $h \in \mathcal{H}_{i+1}$ decreases with each iteration, after enough iterations we will have each $h \in \mathcal{H}_{i+1}$ having $\epsilon$-optimality.

The label complexity bound follows standard ideas in the DBAL literature; see for example [9, 24]. Essentially, what we do is show that at each iteration $i$, because the true error of any $h \in \mathcal{H}_i$ on the multi-group objective can't be too large, the disagreement of $h$ and $h^*$ on any single group cannot be too large. This leads to a bound on the size of the disagreement region for each $g$.

**Theorem 1.** *For all $\epsilon > 0$, $\delta \in (0, 1)$, collections of group distributions $\mathcal{G}$, and hypothesis classes $\mathcal{H}$ with $d < \infty$, with probability $\geq 1 - \delta$, the output $\hat{h}$ of Algorithm 1 satisfies*

$$L_{\mathcal{G}}^{\max}(\hat{h}) \leq L_{\mathcal{G}}^{\max}(h^*) + \epsilon,$$

*and its label complexity is bounded by*

$$\tilde{O}\left( G \, \theta_{\mathcal{G}}^2 \left( \frac{\nu^2}{\epsilon^2} + 1 \right) \left( d \log(1/\epsilon) + \log(1/\delta) \right) \log(1/\epsilon) + \frac{G \log(1/\epsilon) \log(1/\delta)}{\epsilon^2} \right).$$

*Proof.* Lemma 4 says that the number of samples drawn at each iteration is sufficiently large that with probability $\geq 1 - \delta$, for all $i \in [I]$, it holds that for all $h \in \mathcal{H}_{i+1}$, that we have $\left| L_{\mathcal{G}}^{\max}(h) - L_{\mathcal{G}}^{\max}(h^*) \right| \leq 2^{I-i}\epsilon$. Thus, after $I = \lceil \log(1/\epsilon) \rceil$ iterations, the output $\hat{h}$ satisfies the consistency condition.

To see the label complexity, which is the sum of the number of labels we query at each iteration, we note at iteration $i$, we label no more than

$$1024 \left( \frac{m_i}{\epsilon 2^{I-i}} \right)^2 \left( 2d \log\left( \frac{64}{\epsilon} \right) + \ln\left( \frac{8G\lceil \log(1/\epsilon) \rceil}{\delta} \right) \right) + \frac{128 \ln(4G\lceil \log(1/\epsilon) \rceil/\delta)}{\epsilon^2}$$

samples for each group distribution $D_g$, where $m_i = \max_{g'} \mathbb{P}_{D_{g'}}(x \in \Delta(\mathcal{H}_i))$. The only term here that depends on $i$ is $\frac{m_i}{\epsilon 2^{I-i}}$. By Lemma 4, with probability $\geq 1 - \delta$, it holds for each $i > 1$ that $\left| L_{\mathcal{G}}^{\max}(h) - L_{\mathcal{G}}^{\max}(h^*) \right| \leq 2^{I-i+1}\epsilon$; this holds automatically at $i = 1$ by the setting of $I = \lceil \log(1/\epsilon) \rceil$. Thus, at arbitrary $i$ and for arbitrary $g \in [G]$, we may write

$$\begin{aligned}
\rho_g(h, h^*) &= \mathbb{P}_{D_g}(h(x) \neq h^*(x)) \\
&= \mathbb{P}_{D_g}(h(x) \neq y, h^*(x) = y) + \mathbb{P}_{D_g}(h(x) = y, h^*(x) \neq y) \\
&\leq \mathbb{P}_{D_g}(h(x) \neq y) + \mathbb{P}_{D_g}(h^*(x) \neq y) \\
&= L_{\mathcal{G}}(h \mid g) + L_{\mathcal{G}}(h^* \mid g) \\
&\leq L_{\mathcal{G}}^{\max}(h) + L_{\mathcal{G}}^{\max}(h^*) \\
&= L_{\mathcal{G}}^{\max}(h) - L_{\mathcal{G}}^{\max}(h^*) + L_{\mathcal{G}}^{\max}(h^*) + L_{\mathcal{G}}^{\max}(h^*) \\
&\leq 2^{I-i+1}\epsilon + 2\nu,
\end{aligned}$$

where we recall $\nu$ is the noise rate on the multi-group objective. Thus, with probability $\geq 1 - \delta$, for each $i \in I$ and $g \in [G]$, it holds that

$$\mathcal{H}_i \subseteq B_g(h^*, 2^{I-i+1}\epsilon + 2\nu).$$

Given this observation, we may then write, for all $g$, that

$$\mathbb{P}_{D_g}(x \in \Delta(\mathcal{H}_i)) \leq \mathbb{P}_{D_g}(x \in \Delta(B_g(h^*, 2\nu + 2^{I-i+1}\epsilon))),$$

as if there are $h, h' \in \mathcal{H}_i$ that disagree on some $x$, we have $h, h' \in B_g(h^*, 2\nu + 2^{I-i+1}\epsilon)$, and so $h, h'$ also realize disagreement on $x$ for the larger set of classifiers. Recalling the definition of $m_i$, this allows us to bound the sum of terms depending on $i$ for each distribution $D_g$ as

$$
\begin{aligned}
\sum_{i=1}^{I} \left( \frac{m_i}{\epsilon 2^{I-i}} \right)^2 &\leq \sum_{i=1}^{I} \left( \frac{\max_{g'} \mathbb{P}_{D_g}\left(x \in \Delta(B_{g'}(h^*, 2\nu + 2^{I-i+1}\epsilon))\right)}{2^{I-i}\epsilon} \right)^2 \\
&\leq \sum_{i=1}^{I} \left( \max_{g'} \frac{\mathbb{P}_{D_g}\left(x \in \Delta(B_{g'}(h^*, 2\nu + 2^{I-i+1}\epsilon))\right)}{2\nu + 2^{I-i+1}\epsilon} \cdot \frac{2\nu + 2^{I-i+1}\epsilon}{2^{I-i}\epsilon} \right)^2 \\
&\leq 4 \left( \frac{\nu + \epsilon}{\epsilon} \right)^2 \sum_{i=1}^{I} \left( \max_{g'} \frac{\mathbb{P}_{D_g}\left(x \in \Delta(B_{g'}(h^*, 2\nu + 2^{I-i+1}\epsilon))\right)}{2\nu + 2^{I-i+1}\epsilon} \right)^2 \\
&\leq 4 \left( \frac{\nu + \epsilon}{\epsilon} \right)^2 \sum_{i=1}^{I} \left( \max_{g'} \sup_{h \in \mathcal{H}} \sup_{r \geq 2\nu + \epsilon} \frac{\mathbb{P}_{D_g}\left(x \in \Delta(B_{g'}(h, r))\right)}{r} \right)^2 \\
&= 4 \lceil \log(1/\epsilon) \rceil \left( \frac{\nu + \epsilon}{\epsilon} \right)^2 \left( \max_{g'} \theta_{g'} \right)^2 \\
&= 4 \lceil \log(1/\epsilon) \rceil \left( \frac{\nu + \epsilon}{\epsilon} \right)^2 \theta_{\mathcal{G}}^2.
\end{aligned}
$$

The label complexity bound then follows by noting the algorithm labels the same amount of samples for all $G$ groups each iteration, and ignoring the factors of $\log(G)$ and $\log(\log(1/\epsilon))$. $\qquad\square$

## 9.2 Group-Realizable Guarantees

**Theorem 2.** *Suppose Algorithm 2 is run with the active learner $\mathcal{A}_{CAL}$ of [26]. Then for all $\epsilon > 0$, $\delta \in (0, 1)$, hypothesis classes $\mathcal{H}$ with $d < \infty$, and collections of group distributions $\mathcal{G}$ that are group realizable with respect to $\mathcal{H}$, with probability $\geq 1 - \delta$, the output $\hat{h}$ satisfies*

$$L_{\mathcal{G}}^{\max}(\hat{h}) \leq L_{\mathcal{G}}^{\max}(h^*) + \epsilon,$$

*and the number of labels requested is*

$$\tilde{O}\left( dG\theta_{\mathcal{G}} \log(1/\epsilon) \right).$$

*Proof.* The label complexity follows directly from the guarantees given in [15]. By a union bound, we with probability $\geq 1 - \delta$, have that for all $g \in [G]$, that $\mathcal{A}_{CAL}$ returns $\hat{h}_g$ with the property that

$$L_{\mathcal{G}}(\hat{h}_g \mid g) \leq \epsilon/6.$$

Fix some $g \in [G]$ arbitrarily. Consider a counterfactual training set $S_g$, unseen by the learner, constructed by labeling each example $x \in S'_g$ via the oracle call $O_g(x)$. Then Vapnik [28] tells us that $m_g := |S'_g|$ is sufficiently large that with probability $\geq 1 - \delta/2$, for each $h \in \mathcal{H}$ simultaneously, we have

$$\left| L_{\mathcal{G}}(h \mid g) - L_{S_g}(h) \right| < \epsilon/6.$$

Again by the union bound, this uniform convergence on $S_g$ and the guarantee on the runs of $\mathcal{A}_{CAL}$ both hold for each $g \in [G]$. Conditioned on this high probability event, we can first note that for some arbitrary $h \in \mathcal{H}$,

$$
\begin{aligned}
\left| L_{S_g}(h) - L_{\hat{S}_g}(h) \right| &= \left| \frac{1}{m_g} \sum_{i=1}^{m_g} \mathbb{1}[h(x_i) \neq y_i] - \mathbb{1}[h(x_i) \neq \hat{h}_g(x_i)] \right| \\
&\leq \frac{1}{m_g} \sum_{i=1}^{m_g} \left| \mathbb{1}[h(x_i) \neq y_i] - \mathbb{1}[h(x_i) \neq \hat{h}_g(x_i)] \right| \\
&\leq \frac{1}{m_g} \sum_{i=1}^{m_g} \mathbb{1}[y_i \neq \hat{h}_g(x_i)] \\
&= L_{S_g}(\hat{h}_g) \\
&\leq L_{\mathcal{G}}(\hat{h}_g) + \epsilon/6 \\
&\leq \epsilon/6 + \epsilon/6 \\
&= \epsilon/3,
\end{aligned}
$$

where the second to last inequality comes from the uniform convergence over $S_g$, and the final inequality comes from the success of the runs of $\mathcal{A}_{CAL}$. Then for arbitrary $h$, combining Vapnik's guarantee and the inequality we just showed, we may write:

$$
\begin{aligned}
\left| L_{\mathcal{G}}(h \mid g) - L_{\hat{S}_g}(h) \right| &= \left| L_{\mathcal{G}}(h \mid g) - L_{S_g}(h) + L_{S_g}(h) - L_{\hat{S}_g}(h) \right| \\
&\leq \left| L_{\mathcal{G}}(h \mid g) - L_{S_g}(h) \right| + \left| L_{S_g}(h) - L_{\hat{S}_g}(h) \right| \\
&< \epsilon/6 + \epsilon/3 \\
&= \epsilon/2.
\end{aligned}
$$

Given this guarantee on the representativeness of the artificially labeled samples on each group $g$, we have a guarantee for the representativeness over the worst case. For arbitrarily $h \in \mathcal{H}$, we may write

$$
\begin{aligned}
\left| L_{\mathcal{G}}^{\max}(h) - \max_{g \in [G]} L_{\hat{S}_g}(h) \right| &= \left| \max_{g \in [G]} L_{\mathcal{G}}(h \mid g) - \max_{g \in [G]} L_{\hat{S}_g}(h) \right| \\
&\leq \max_{g \in [G]} \left| L_{\mathcal{G}}(h \mid g) - L_{\hat{S}_g}(h) \right| \\
&\leq \epsilon/2.
\end{aligned}
$$

Thus, by the fact that $\hat{h}$ is the ERM on the artificially labeled dataset, we have

$$
L_{\mathcal{G}}^{\max}(\hat{h}) \leq \max_{g \in [G]} L_{\hat{S}_g}(\hat{h}) + \epsilon/2 \leq \max_{g \in [G]} L_{\hat{S}_g}(h^*) + \epsilon/2 \leq L_{\mathcal{G}}^{\max}(h^*) + \epsilon.
$$

$\square$

### 9.3 Approximation Guarantees

**Theorem 3.** *Suppose Algorithm 3 is run with the active learner $\mathcal{A}_{DHM}$ of [15]. Then for all $\epsilon > 0$, $\delta \in (0, 1)$, hypothesis classes $\mathcal{H}$ with $d < \infty$, and collections of groups $\mathcal{D}$, with probability $\geq 1 - \delta$, the output $\hat{h}$ satisfies*

$$
L_{\mathcal{G}}^{\max}(\hat{h}) \leq L_{\mathcal{G}}^{\max}(h^*) + 2 \cdot \max_{g \in [G]} \nu_g + \epsilon \leq 3 \cdot L_{\mathcal{G}}^{\max}(h^*) + \epsilon,
$$

*and the number of labels requested is*

$$
\tilde{O}\left( dG\theta_{\mathcal{G}}\left( \log^2(1/\epsilon) + \frac{\nu^2}{\epsilon^2} \right) \right).
$$

*Proof.* The proof is almost identical to that of Theorem 2. The label complexity bound follows directly from [10]. Similar to before, we have that for all $g \in [G]$, $\mathcal{A}_{DHM}$ returns $\hat{h}_g$ with the property that

$$L_{\mathcal{G}}(\hat{h}_g \mid g) \le L_{\mathcal{G}}(h_g^* \mid g) + \epsilon/6.$$

Fix some $g \in [G]$ arbitrarily. On a counterfactual training set $S_g$, unseen by the learner, constructed by labeling each example $x \in S_g'$ via the oracle call $O_g(x)$, it holds that $m_g := |S_g'|$ is sufficiently large that with probability $\ge 1 - \delta/2$, for each $h \in \mathcal{H}$ simultaneously, we have

$$\left| L_{\mathcal{G}}(h \mid g) - L_{S_g}(h) \right| < \epsilon/6.$$

By the union bound, this uniform convergence and the guarantee on the runs of $\mathcal{A}_{DHM}$ both hold. Thus, we can first note that for some arbitrary $h \in \mathcal{H}$,

$$
\begin{aligned}
\left| L_{S_g}(h) - L_{\hat{S}_g}(h) \right| &= \left| \frac{1}{m_g} \sum_{i=1}^{m_g} \mathbb{1}[h(x_i) \ne y_i] - \mathbb{1}[h(x_i) \ne \hat{h}_g(x_i)] \right| \\
&\le \frac{1}{m_g} \sum_{i=1}^{m_g} \left| \mathbb{1}[h(x_i) \ne y_i] - \mathbb{1}[h(x_i) \ne \hat{h}_g(x_i)] \right| \\
&\le \frac{1}{m_g} \sum_{i=1}^{m_g} \mathbb{1}[y_i \ne \hat{h}_g(x_i)] \\
&\le L_{\mathcal{G}}(\hat{h}_g \mid g) + \epsilon/6 \\
&\le L_{\mathcal{G}}(h_g^* \mid g) + \epsilon/3 \\
&= \nu_g + \epsilon/3.
\end{aligned}
$$

where the second to last inequality comes from uniform convergence over $S_G$, and the final equality comes from the correctness guarantee of $\mathcal{A}_{DHM}$. Then for arbitrary $h$, combining Vapnik's guarantee and the inequality we just showed, we may write:

$$
\begin{aligned}
\left| L_{\mathcal{G}}(h \mid g) - L_{\hat{S}_g}(h) \right| &= \left| L_{\mathcal{G}}(h \mid g) - L_{S_g}(h) + L_{S_g}(h) - L_{\hat{S}_g}(h) \right| \\
&\le \left| L_{\mathcal{G}}(h \mid g) - L_{S_g}(h) \right| + \left| L_{S_g}(h) - L_{\hat{S}_g}(h) \right| \\
&< \epsilon/6 + \nu_g + \epsilon/3 \\
&= \nu_g + \epsilon/2.
\end{aligned}
$$

Then, as above, we have, for arbitrarily $h \in \mathcal{H}$,

$$\left| L_{\mathcal{G}}^{\max}(h) - \max_{g \in [G]} L_{\hat{S}_g}(h) \right| \le \max_{g \in [G]} \left| L_{\mathcal{G}}(h \mid g) - L_{\hat{S}_g}(h) \right| \le \max_{g \in [G]} \nu_g + \epsilon/2 \le \nu + \epsilon/2,$$

where the the final inequality comes from the fact that if any hypothesis has less than $\nu_g$ error on all groups, it would be optimal on group $g$. Thus, by the fact that $\hat{h}$ is the ERM on the artificially labeled dataset, we have

$$L_{\mathcal{G}}^{\max}(\hat{h}) \le \max_{g \in [G]} L_{\hat{S}_g}(\hat{h}) + \nu_g + \epsilon/2 \le \max_{g \in [G]} L_{\hat{S}_g}(h^*) + \nu_g + \epsilon/2 \le L_{\mathcal{G}}^{\max}(h^*) + 2\nu + \epsilon \le 3 \cdot L_{\mathcal{G}}^{\max}(h^*) + \epsilon$$

$\square$

