# OpenReview forum: "Agnostic Multi-Group Active Learning"
_NeurIPS.cc/2023/Conference — NeurIPS 2023 poster_

### Official Review · Reviewer_T3gQ · 2023-06-22

**Soundness:** 2 fair
**Presentation:** 3 good
**Contribution:** 2 fair
**Rating:** 4
**Confidence:** 3

**Summary:**

This paper studies active learning in a multi-group setting, in which the learner performs adaptive sampling over groups of distributions for solving a binary classification task. The performance of the learner is measured by its maximal loss over the groups. The main goal is to extend the disagreement-based active learning approaches to the multi-group setting. To achieve this, the paper proposes a novel worst-case (over groups) loss estimator for the learner. The theoretical analyses on the sample complexity of the algorithms using this new loss estimator are provided.

**Strengths:**

- The paper considers the multi-group extension of the active learning problem, which is likely a hard problem of high interest for the active learning community. The setting considered in the problem can also be seen as a variant of the Group-DRO problem in which the set of hypotheses and the loss function are not assumed to be convex.
- The paper shows rigorously why the traditional disagree-based active learning approaches in the multi-group setting, and proposes a novel multi-group loss estimator with reasonable theoretical guarantees.
- For the most parts, the writing is clear and easy to follow.

**Weaknesses:**

Major:
1. The main algorithm, Algorithm 1, seems computationally intractable. A thorough computational complexity analysis is lacking, especially in the most important steps in Algorithm 1 on line 4 (computing $R_i$), line 21 (computing $\hat{h}_i$) and line 22. These steps involve filtering two sets ($\mathcal{X}$ and $\mathcal{H}_i$) that are possibly uncountably infinite, however no explanation is given on how these steps are done. This reduces the impact of the proposed approach to practical scenarios.
2. It is not clear why the sample complexity bound given in Theorem 1 is an improvement over passive and collaborative learning. The bound in Theorem 1 still depends on $d/\epsilon^2$, which is confusing since the text wrote ``the interaction between $d$ and $1/\epsilon^2$ is removed". In fact, the additional multiplicative factor $G\theta_g^2\log(1/\epsilon)\nu$ can be larger than 1, which makes the bound in Theorem 1 worse than the collaborative learning bound shown in Table 1.
3. A lower bound is lacking, which makes it impossible to judge how tight the upper bounds are.
4. Empirical evaluation is lacking.

Minor:
1. Throughout the paper, the references to various technical lemmas are missing. In Section 5.2, the paper wrote "it is a lemma of ours that...", with no references. Similarly, the proof of Lemma 1 in the Appendix (line 433) mentions "a lemma of Vapnik [28]", which is meaningless because Vapnik [28] is a 700-page book.
2. One the same line 433 - 434, the $1/m$ term in the uniform convergence bound looks redundant. Mathematically it might be correct, but a tighter (and simpler) bound that only depends on $\sqrt{\frac{\ln(8/\delta) + d\ln(2em/d)}{m}}$ (up to constants) are well-known and can be found in any standard machine learning texts (e.g. see [1], section 28.1, page 393).
3. This factor of $1/m$ is missing in the second equation on line 440, the first term.
4. In the main text: "with the ability decide" -> with the ability to decide; "not enough infer" -> not enough to infer; "would chose" -> would choose.
5. Typo in the equations in Example 1: it should be $\ell(h, i, A)$, not $\ell(h, i, S)$.

[1] Shai Shalev-Shwartz and Shai Ben-David. 2014. Understanding Machine Learning: From Theory to Algorithms. Cambridge University Press, USA.


**Questions:**

1. Can the authors comment on the computational complexity issue above regarding the computation of $R_i$ and $\mathcal{H}_{i+1}$?
2. How exactly is the bound in Theorem 1 better than the existing bounds in related settings?
3. How can we compare the results in the paper with the sample complexity lower bound for adaptive sampling strategies in the Group-DRO setting [1]? I am aware that the two settings have different assumptions on the convexity and compactness of $\mathcal{H}$ and  $\ell$, so my question is restricted to compact convex $\mathcal{H}$ and convex $\ell$ only. In other words, would the lower bound in [1] apply if $\mathcal{H}$ is compact convex and $\ell$ is convex in your active learning setting?

[1] Optimal algorithms for group distributionally robust optimization and beyond. Soma et al., 2022.

**Limitations:**

I suggest the authors to include more detailed discussion of the limitations of the proposed algorithms in the main text. These include any computational as well as theoretical issues raised above.

---

> ### Author Rebuttal · Authors · 2023-08-09
>
> We thank the reviewer for the comments, and the close reading even of parts of our Appendix.
>
> >Can the authors comment on the computational complexity issue above regarding the computation of $R_i$ and $\mathcal{H}_{i+1}$?
>
> With regard to computational considerations, one can always construct an epsilon cover of the hypothesis (of near-optimal size using a polynomial number of unlabeled examples [2]) and reduce to learning over a finite class, in which case one can directly execute the algorithm.
>
> > It is not clear why the sample complexity bound given in Theorem 1 is an improvement over passive and collaborative learning. The bound in Theorem 1 still depends on, which is confusing since the text wrote ``the interaction between and is removed".....How exactly is the bound in Theorem 1 better than the existing bounds in related settings?
>
> The reviewer is correct that the interaction between $d$ and $1/\epsilon^2$ is not 'removed' unless the noise rate is 0; this is a writing error that we will fix in the next version. That said, as we discuss in Section 5.3, when the noise rate is small enough, it will be dominated by other terms. This is exactly the type of guarantee one finds in canonical 0-1 loss active learning literature [11]. As we also note in Section 5.3, we also need the number of groups to be small to improve over collaborative learning lower bounds.
>
> It should further be mentioned that the upper bounds in collaborative learning are on the expectation over a draw from a distribution over the hypothesis class [1], which is a very different style of guarantee than what we present here.
>
> > How can we compare the results in the paper with the sample complexity lower bound for adaptive sampling strategies in the Group-DRO setting?
>
> This is an interesting question that has pointed us to some cool related work by Soma et al. [12] that we will cite in the next version of our paper. While a complete mathematical comparison would require new algorithmic analysis on our part, the game-theoretic approach taken by [12] leads to a reduced dependence number of groups which we do not achieve in this paper; this, much like in the comparison to the collaborative learning lower bounds, leads us to the requirement that the number of groups be small for tighter guarantees. We do note that our algorithm is designed to work in a more general setting where convexity assumptions cannot be exploited, so it's not entirely unreasonable that our guarantees do not match theirs in this special case.
>
> &nbsp;
>
> [1] On-Demand Sampling: Learning Optimally from Multiple Distributions. Haghtalab et al., 2022.
> [2] A Bound on the Label Complexity of Agnostic Active Learning. Hanneke 2007. \
> [3] Agnostic Active Learning. Balcan et al., 2006. \
> [4] A general agnostic active learning algorithm. Dasgupta et al., 2007
> [11] The two faces of active learning. Dasgupta, 2012.
> [12] Optimal algorithms for group distributionally robust optimization and beyond. Soma et al., 2022.

---

> > ### Comment · Reviewer_T3gQ · 2023-08-18
> >
> > I have read the authors' responses. In Section 3.1, the paper states that the strong assumption $\nu = 0$ is not used, and this seems to be an advantage of the paper over existing works. However, the rest of paper (and also in the authors' rebuttals) indicates that the bound in Theorem 1 (the main theorem of the paper) is either significant and/or comparable to existing bounds only when $\nu$ is small and close to zero. This seems to be a major contradiction that has not been properly addressed in the rebuttals, hence my score remains unchanged.

---

> > > ### Author Response · Authors · 2023-08-19
> > >
> > > The lack of assumption on $\nu$ presents the challenge of creating a statistically consistent algorithm in arbitrary noise regimes, which this paper solves.
> > >
> > > It is known in the active learning literature that active algorithms only improve over passive approaches in low noise regimes: in the case of the 0-1 loss, one has a label complexity lower bound of $\Omega(d\nu^2/\epsilon^2)$ [12]. Thus, any active scheme one designs will have this term of $d \nu^2/\epsilon^2$ in the upper bound, and improvement over passive will rely on $\nu$ being small.
> > >
> > > Conceptually, active learning algorithms thus give the following advantage over passive: if one believes that $\nu$ is small, but cannot be sure, running the active scheme will definitely get you an optimal classifier in the limit - no matter what the noise - but you will only enjoy a label complexity speed up if you're right about the noise rate $\nu$ being small. Our work presents an algorithm with such guarantees.
> > >
> > > &nbsp;
> > >
> > > The two faces of active learning. Dasgupta, 2012. [12]

---

### Official Review · Reviewer_7xXY · 2023-07-05

**Soundness:** 3 good
**Presentation:** 4 excellent
**Contribution:** 3 good
**Rating:** 5
**Confidence:** 4

**Summary:**

This paper looks at disagreement-based active learning (DBAL) where
the goal is to minimize the maximum error over a (known) set of G
distributions ("groups").  One could learn each group individually,
but the optimum for each group may be very bad on other groups.  By
combining the search, a decent sample complexity can be obtained.


**Strengths:**

It's an interesting problem, fairly well motivated and it's clearly
explained (Example 1) why standard DBAL won't quite cut it.  You
definitely need to do something clever to handle having multiple
groups, and the bound they get is competent.

I enjoyed reading the paper.

**Weaknesses:**

The main problem with Theorem 1 is that the result has theta^2 where
standard DBAL gets theta (here, theta is the disagreement
coefficient).  It really seems like theta should be possible here.

In the easier regime of getting error O(eta) [Theorem 3], the
dependence is theta as one would expect.  But the bound there is d G
theta log^2(1/eps), and it seems like the log^2 should be log(1/eps):
Standard DBAL only needs log(1/eps).  Something like:

(1) get loss (eta + eps) within each class, using d G theta log
(1/eps) samples.  This gives hypotheses h_1, ..., h_G such that each
h_g has error at most (eta + eps) on class g.

(2) output the hypothesis h^ that minimizes max_g Pr_{x ~ g} [ h^ != h_g].

Since the optimum h* has error eta on each class relative to the truth on class g,
it has error at most (2 eta + eps) relative to h_g
so h^ has error at most (2 eta + eps) relative to h_g on g
so h^ has error at most (4 eta + 2 eps) relative to h* on g.

I guess this is a 4 eta approximation, not a 3 eta approximation, but
that doesn't seem very significant (also unsure if it's necessary).



**Questions:**

Is there any reason to expect the theta^2 or log^2 dependences in Theorem 1/3 to be right?

Is it possible to have experimental evidence that this stuff ever works, even synthetically?

**Limitations:**

Yes.

---

> ### Author Rebuttal · Authors · 2023-08-09
>
> We thank the reviewer for reading our work with an eye for algorithmic detail.
>
> > In the easier regime of getting error $O(\epsilon)$ [Theorem 3], the dependence is theta as one would expect. But the bound there is $d G \theta \log^2(1/\epsilon)$, and it seems like the $\log^2(1/\epsilon)$ should be $\log(1/\epsilon)$.
>
> This label complexity is inherited directly from the choice of the 0-1 loss active learner due to [10] that we instantiated to simplify the theorem statement. In accordance with the algorithmic ideas outlined by the reviewer, the label complexity in Theorem 3 is G * ('pick your favorite 0-1 loss agnostic active learners label complexity'), and so by feeding a better active learner to Algorithm 2, you can automatically shave down the dependence on $\epsilon$ to below $\log(1/\epsilon)^2$ without any extra analysis.
>
> > The main problem with Theorem 1 is that the result has $\theta^2$ where standard DBAL gets theta (here, theta is the disagreement coefficient). It really seems like theta should be possible here.
>
> As the reviewer points out, the dependence on $\theta^2$ in Theorem 1 is likely unnecessary.  Our understanding of the history of the agnostic active learning literature is that the first general analysis of agnostic active strategies from Hanneke [2] showed this $\theta^2$ factor, and that through better concentration bounds and more subtle analysis, the extra $\theta$ was shown to be unnecessary [8]. Basically, we did not set out to fully optimize the bounds given the algorithmic idea of Algorithm 1, instead opting for a more straightforward analysis to show the algorithmic idea is viable.
>
> We will take a closer look to see if we can improve the bounds. This usually comes from using concentration inequalities that take into account the variance of the estimator, as in Section 5 of [15].
>
> &nbsp;
>
> [2] A Bound on the Label Complexity of Agnostic Active Learning. Hanneke, 2007. \
> [8] Theory of Active Learning. Hanneke, 2014. \
> [9] Agnostic Active Learning Without Constraints. Beygelzimer et al., 2010. \
> [10] A general agnostic active learning algorithm. Dasgupta et al., 2007. ' \
> [15] Theory of Classification: A Survey of Some Recent Advances. Boucheron, 2005.

---

> > ### Comment · Reviewer_7xXY · 2023-08-12
> >
> > Thanks for your response.  I guess that's fair, and I'll raise my score to 5.

---

### Official Review · Reviewer_rZgu · 2023-07-07

**Soundness:** 4 excellent
**Presentation:** 4 excellent
**Contribution:** 3 good
**Rating:** 7
**Confidence:** 3

**Summary:**

The paper considers an active binary classification setting where the feature space is partitioned into several groups. The performance of the learner is given as their worst case performance across all groups. The effect of considering such a multi-group setting is that, it is no longer efficient to sample exclusively in the disagreement region of said class. The authors propose an algorithm which iteratively eliminates hypothesis from the hypothesis class, while keeping track of the performance of hypothesis outside the disagreement area. The trick they use, is to split the loss of a classifier between the loss on the disagreement region and its complement. The later can be estimated by the loss of an arbitrary hypothesis from all remaining hypothesis. This removes dependence on the VC dimension from the estimation of the second term. They give PAC guarantees for their algorithm with an upper bound on expected sampling time.

The authors also consider the problem of group realizable learning. In this setting a perfect classifier is assumed to exist for each group. Here, they propose an algorithm which first runs a sub routine to find an $\epsilon$ good classifier on each group. They then use said classifiers to label points in each group and finally do ERM with respect to the multi group objective. They provide PAC guarantees for the algorithm with an upper bound on the sampling time.

**Strengths:**

The paper is well written and easy to read.The motivating example of section 4.2 gives good intuition as to why it is not sufficient to sample on in the disagreement region, which is in itself an interesting phenomenon. The mechanism of algorithm 1 is well described and the reader is given good intuition as to how it's sampling strategy leads to a reduction in VC dimension.

**Weaknesses:**

The authors compare the result of theorem 1 to a lower bound from passive learning. It would be of more interest to consider lower bounds in the active setting, even in the form of conjectures. Discussion on the necessity of the $\log(1/\epsilon)$ term is also missing.

**Questions:**

Have the authors considered a setting where, instead of two oracles, the learner can only receive limited samples from the Rademacher random variable with parameter $\mathbb{P}_{D_g}(Y=1 | x)$, for an $x$ of there choosing, where the partition of the feature space across the groups is known. In particular this should change the nature of group realizable learning.

**Limitations:**

The authors could address the lack of lower bounds for the active setting in more detail. Also Theorem 3 is presented in isolation, it would be nice to have some speculation on whether this result is optimal and also to view it in context of previous literature.

---

> ### Author Rebuttal · Authors · 2023-08-09
>
> We thank the reviewer for a thorough reading of our submission. We try to address some questions/concerns below.
>
> >It would be of more interest to consider lower bounds in the active setting, even in the form of conjectures
>
> We agree it would be nice to have active lower bounds. In general, tight lower bounds in a disagreement context can’t be expected, in that there is no literature saying that the relics of disagreement based methods are necessary (read: disagreement coefficient). Partially for this reason, it's fairly common in active learning that you see comparisons to passive lower bounds. That said, some lower bound is achievable.
>
> >Have the authors considered a setting where, instead of two oracles, the learner can only receive limited samples from the Rademacher random variable with parameter $\mathbb{P}_{D_G}(Y= 1|x)$, for an $x$ of there choosing
>
> The alternate oracle/sampling model suggested by the reviewer seems to be that of  ''membership query'' active learning [5]. There are some interesting results in this field [6], but in general, this query model has gone out of favor in the AL literature, because in practice the algorithms tend to ask for labels of $x$ that are unnaturally close to the decision boundary, e.g. strange mixtures of cat and dog. It might be an interesting problem to consider though, especially in combination with some minor modifications to make queries more realistic (see [7]).
>
> &nbsp;
>
> [5] Queries and Concept Learning, Angluin, 1988. \
> [6] An Efficient Membership-Query Algorithm for Learning DNF with Respect to the Uniform Distribution, Jackson, 1996. \
> [7] Learning using Local Membership Queries under Smooth Distributions, Awasthi et al., 2012.

---

> > ### Comment · Reviewer_rZgu · 2023-08-17
> >
> > Thank you for the detailed answer to my question, that makes sense! My score remains unchanged.

---

### Official Review · Reviewer_eXQR · 2023-07-07

**Soundness:** 3 good
**Presentation:** 3 good
**Contribution:** 3 good
**Rating:** 6
**Confidence:** 3

**Summary:**

This paper studies multi-group learning from the active learning perspective. The authors first discuss why classical disagreement-based active learning methods cannot work and provide an example. Then, they propose a modified algorithm and provide theoretical guarantees. The label complexity bound is smaller in order than the lower bound in collaborative learning under certain condition. For the special case of group realizable setting, they also establish improved theoretical results.

**Strengths:**

1. This paper is well-written and easy to follow.
2. This paper clearly states why the classical disagreement-based active learning methods cannot work in the multi-group learning setting. The example in Section 4.2 is interesting.
3. The authors compare their label complexity bound with the lower bound in collaborative learning setting and show that their bound is smaller in order under certain condition.
4. Improved theoretical results are achieved for the group realizable setting.

**Weaknesses:**

1. This paper only discusses the drawback of disagreement-based methods in multi-group learning. How about uncertainty-based methods? What if the learner just label the sample from the most uncertain group?
2. From line 22 in algorithm 1, the proposed method is computationally inefficient and cannot be implemented in practice.

Minor:
The caption of table 1 states that $\widetilde{\mathcal{O}}$ hides $\log(1/\epsilon)$, it should be $\log(\log(1/\epsilon))$?

**Questions:**

1. How about uncertainty-based methods? What if the learner just label the sample from the most uncertain group?
2. Is the proposed algorithm computationally efficient? Are there any insights for the computational complexity?

---

> ### Author Rebuttal · Authors · 2023-08-09
>
> We appreciate the review and the comments.
>
> > How about uncertainty-based methods? What if the learner just labels the sample from the most uncertain group?
>
> In general, one must be careful with uncertainty based approaches in the agnostic AL setting as they can easily lead to inconsistent strategies (the issue of consistency of active learning strategies is described concisely in [11]).
>
> Further, not discriminating the labeling beyond the group-level will not yield better guarantees than the lower bounds of [1], as this is the sampling regime of so-called 'collaborative learning'. The issue with directly combining ideas in collaborative learning[1] with those given in our algorithm is that upper bounds in this literature are on an expected error rate over a draw from a distribution over the hypothesis class, which is a weaker sort of guarantee that isn’t easy to exploit when one wants to output a deterministic hypothesis.
>
> > From line 22 in algorithm 1, the proposed method is computationally inefficient and cannot be implemented in practice.... Is the proposed algorithm computationally efficient?
>
> As we mention in the Author Rebuttal, one can always construct an epsilon cover of $\mathcal{H}$ of near-optimal size using a polynomial number of unlabeled samples [2]. This will yield a finite class on which one can execute the algorithm while still maintaining guarantees. There are some techniques in the literature for making active strategies directly executable on infinite classes [4, 14], but we leave the application of such ideas to our problem for future work.
>
> >Minor: The caption of table 1 states that $\tilde{O}$ hides $\log(1/\epsilon)$,  it should be $\log(\log(1/\epsilon))$?
>
> The caption in Table 1 is correct, if strangely worded. The $\tilde{O}$ hides factors of $\log(\log(1/\epsilon))$, i.e. which we write as 'factors logarithmic in $\log(1/\epsilon)$'. We will update this.
>
> &nbsp;
>
> [1] On-Demand Sampling: Learning Optimally from Multiple Distributions. Haghtalab et al., 2022. \
> [2] A Bound on the Label Complexity of Agnostic Active Learning. Hanneke 2007. \
> [3] Agnostic Active Learning. Balcan et al., 2006. \
> [4] A general agnostic active learning algorithm. Dasgupta et al., 2007
> [11] The two faces of active learning. Dasgupta, 2012.
> [14] Agnostic Active Learning Without Constraints, Beygelzimer et al., 2010.

---

> > ### Comment · Reviewer_eXQR · 2023-08-21
> >
> > Thanks for your response. I will keep my score.

---

### Author Rebuttal · Authors · 2023-08-09

We appreciate all the reviews.

There were a few comments on the fact that the general agnostic algorithm (Algorithm 1), as stated, cannot be explicitly executed for infinite hypothesis classes.

In the case that one wishes to run the algorithm, one can always construct an epsilon-cover of the hypothesis class $\mathcal{H}$ (of near-optimal size using a polynomial number of unlabeled samples [2]), and then execute the algorithm on the resulting finite class, while maintaining guarantees.

Beyond this, prior work in active learning has come up with algorithmic techniques, such as leave-one-out training [14] and constrained optimization [4], for making active learning strategies directly computationally feasible on uncountable hypothesis classes. How to apply those for this specific problem is an interesting direction of future work, something we will discuss in an updated version of the paper.

&nbsp;

[2] A Bound on the Label Complexity of Agnostic Active Learning. Hanneke 2007. \
[4] A general agnostic active learning algorithm. Dasgupta et al., 2007 \
[14] Agnostic Active Learning Without Constraints, Beygelzimer et al., 2010.

---

### Decision · Program_Chairs · 2023-09-21

**Decision:**

Accept (poster)

**Comment:**

The paper studies the classical setting of active learning but in the multi group setting. This is an important setting given applications to fairness and robustness. The authors show why standard disagreement based active learning approaches fail and propose modifications under which one can obtain label complexity upper bounds. Overall the reviewers are supportive of the paper. However, the authors are encouraged to take the reviewer comments into account in order to improve the paper especially connections to recent work on collaborative learning.